# Laboratory Test Results in Patients with Workplace Moisture Damage Associated Symptoms—The SAMDAW Study

**DOI:** 10.3390/healthcare11070971

**Published:** 2023-03-29

**Authors:** Pia Nynäs, Sarkku Vilpas, Elina Kankare, Jussi Karjalainen, Lauri Lehtimäki, Jura Numminen, Antti Tikkakoski, Leenamaija Kleemola, Jukka Uitti

**Affiliations:** 1Faculty of Medicine and Health Technology, Tampere University, 33520 Tampere, Finland; 2Department of Phoniatrics, Tampere University Hospital, 33520 Tampere, Finland; 3Allergy Centre, Tampere University Hospital, 33520 Tampere, Finland; 4Department of Clinical Physiology and Nuclear Medicine, Tampere University Hospital, 33520 Tampere, Finland

**Keywords:** moisture damage, mold, dampness, blood count, CRP, FeNO, IgE, allergy, multiple chemical sensitivity

## Abstract

The mechanisms of health effects of moisture damage (MD) are unclear, but inflammatory responses have been suspected. The usefulness of laboratory and allergy tests among patients in secondary healthcare with symptoms associated with workplace MD were examined. Full blood count, erythrocyte sedimentation rate (ESR), C-reactive protein (CRP), total serum immunoglobulin E (IgE), fractional exhaled nitric oxide (FeNO), and skin prick testing were assessed and analyzed in relation to multiple chemical sensitivity (MCS) and perceived stress in 99 patients and 48 controls. In analysis, *t*-tests, Mann-Whitney tests, and chi-squared tests were used. Minor clinically insignificant differences in blood counts were seen in patients and controls, but among patients with asthma an elevated neutrophil count was found in 19% with and only in 2% of patients without asthma (*p* = 0.003). CRP levels and ESR were low, and the study patients’ FeNO, total IgE, or allergic sensitization were not increased compared to controls. The level of stress was high among 26% of patients and 6% of controls (*p* = 0.005), and MCS was more common among patients (39% vs. 10%, *p* < 0.001). Stress or MCS were not significantly associated with laboratory test results. In conclusion, no basic laboratory or allergy test results were characteristic of this patient group, and neither inflammatory processes nor allergic sensitization were found to explain the symptoms among these patients. While the value of basic laboratory tests should not be ignored, the use of allergy tests does not seem necessary when symptoms are indicated to be workplace-related.

## 1. Introduction

Various health effects, such as upper respiratory tract symptoms and the development or deterioration of asthma, have all been associated with exposure to moisture damage (MD) at the workplace [1,2]. The possible mechanisms of health effects associated with MD are unclear but based on findings in some studies, inflammatory responses to MD exposure have been suspected [3]. To prove the development of inflammatory responses studies have explored the levels of C-reactive protein (CRP), blood leucocytes and eosinophils (B-eos), serum total immunoglobulin E (IgE), and fractional exhaled nitric oxide (FeNO) in subjects with MD exposure in different indoor environments. In a study among 6-year-old children with confirmed MD at home, there was a significant positive association between MD exposure and increased serum CRP level. This was not observed with blood leucocyte or FeNO levels [4]. In a ten-year follow-up study among Swedish adults, serum total IgE and CRP levels were found to be predictors of MD-associated symptoms in homes. In the follow-up, smoking decreased, and self-reported hay fever increased significantly while reports of moisture damage at homes somewhat diminished [5].

It must be noted that exposure to possible MD in the workplace differs from that at home: less time is usually spent in the workplace, and the premises, indoor air conditions, especially ventilation, and activities are likewise different. A cross-sectional study using the level of fungal DNA in settled dust as a workplace MD marker revealed elevated levels of CRP and FeNO in employees of Swedish daycare centers with higher fungal DNA levels at the workplace [6]. In another Swedish study with a ten-year follow-up, B-eos was associated with workplace MD at baseline in a random sample of subjects. In the follow-up, MD (self-reported signs of dampness and/or mold odor) was associated with increased symptoms but not with the levels of CRP or serum total IgE [7]. Similar results were obtained in a study with repeated blood samples over 14 months that found no difference in CRP or IgE levels between personnel of MDd and control buildings [8]. 

Multiple chemical sensitivity (MCS) is a condition in which an individual develops symptoms in different organ systems related to low-level chemical exposure that is not known to cause health effects and does not usually cause symptoms in people [9]. MCS is a subtype of environmental intolerance (EI) [10] which includes reacting to different environmental factors such as chemicals or odors. EI can also be building-related [11]. EI symptoms cannot be explained by any known toxicological [12], physical [13], or immunological [14], mechanisms [15]. Recent studies suggest that the key mechanisms causing EI could be central sensitization and change in the neurological processing of sensory stimuli [11,16,17]. The development of MCS has been linked with perceived stress [18] which on the other hand has been found to associate with both indoor air-associated symptoms [19] and inflammatory responses [20]. As there is no recognized biological mechanism explaining MCS, there are no clinical tests for the diagnosis. To screen the presence of MCS, different questionnaires have been developed of which The Quick Environmental Exposure and Sensitivity Inventory (QEESI©) [21] seems to be the most widely used. 

The existing guidelines [22,23] to examine patients with MD-associated symptoms are mainly based on studies of the health effects of MD exposure at home, and most of these studies have focused on children. According to German-Austrian guidelines, doctors are encouraged to assess family medical history regarding allergies even if the significance of predisposition to allergies in MD-associated symptoms is unclear [22]. Routine laboratory and allergy tests are often used when assessing patients with symptoms associated with workplace MD in health care. However, there was and remains a lack of clinical research on patients with workplace MD exposure-associated symptoms. In this study, the aim was to examine the usefulness of laboratory and allergy tests among these patients as a part of their comprehensive clinical evaluation. In addition, associations between MCS and work-related stress and laboratory test findings were investigated.

## 2. Materials and Methods

Patients who were referred to the Tampere University Hospital departments of Occupational Medicine, Phoniatrics, or the Allergy Centre for evaluation of respiratory or voice symptoms associated with MD exposure at the workplace were recruited to the study. The study inclusion criteria were (1) age between 18 and 65 years, (2) upper and/or lower respiratory tract and/or voice symptoms that are associated with the workplace, and (3) strong suspicion or evidence of MD at the workplace. The exclusion criteria were (1) severe illness (e.g., cancer) and (2) pregnancy. Comprehensive tests and clinical examinations were previously conducted to diagnose possible asthma and chronic rhinosinusitis (CRS). Laboratory tests included single analyses of full blood count, erythrocyte sedimentation rate (ESR), CRP, total serum IgE, and FeNO. Since smoking may cause elevated leucocyte levels [24] and decrease FeNO [25], total leucocyte count (TLC) was analyzed separately in non-smokers and FeNO was omitted from smokers’ testing. Skin prick testing (SPT) was conducted using common allergen extracts (birch, timothy, mugwort, horse, dog, cat, house dust mite *Dermatophagoides pteronyssinus*, latex, and *Aspergillus fumigatus*) (Soluprick, ALK A/S, Copenhagen, Denmark). These were carried out by trained nurses according to a standardized protocol [26]. The SPT was considered positive for allergic sensitization if the wheal size was at least 3 mm larger than the negative control (saline). 

In addition, each patient filled out a questionnaire including QEESI© which has been developed for use in research as well as a clinical evaluation of MCS. Three QEESI© subscales were used to assess possible MCS: the chemical intolerance subscale to identify which chemicals or odors are suspected to cause symptoms, the symptom severity subscale to examine the nature and severity of symptoms a person commonly experiences, and the life impact subscale to assess how the sensitivities affect different aspects of everyday life. The respondents rated each item in different subscales between 0 and 10 points, 0 meaning not at all a problem and 10 severe or disabling problems. The points of each subscale were tallied to obtain a total score from 0 to 100. A high score (40–100 points) in the chemical intolerance subscale was used as a criterion for MCS [21]. 

Work-related stress was assessed with a validated single-item question “Stress means a situation in which a person feels tense, restless, nervous, or anxious or is unable to sleep at night because his/her mind is troubled all the time. Do you feel this kind of stress these days?” by using a 5-point Likert scale from 0 “not at all” to 4 “very much”. The responses were dichotomized into a low-stress level (responses from 0 to 2) or a high-stress level (3 and 4) [27]. 

Symptomless subjects with similar proportions of women and men in different age groups as in the study population were recruited as controls. Except for the absence of the CRP measurement, the controls were subject to the same laboratory tests as the patients.

To compare continuous variables, the independent samples *t*-test and Mann-Whitney test were used. The distributions of the parameters were analyzed from descriptives (differences between mean and 5% trimmed mean, skewness), Q-Q plots, and histograms. For categorical variables, the chi-squared test was used. Data management and analysis were performed using IBM^®^ SPSS^®^ Statistics Version 28 (2021).

The Ethics Committee of the Pirkanmaa Hospital District approved the study (R14095), and all the study participants gave written informed consent.

## 3. Results

The study population consisted of 99 patients, of whom 82 (83%) were women and 17 (17%) were men. Their age varied between 20 and 63 years (mean 44 years). Of the 48 controls, 37 (77%) were women and 11 (23%) were men, their ages varying between 21 and 60 (mean 44) years. 

### 3.1. Laboratory Test Results

#### 3.1.1. Blood Count

The blood count results were normally distributed both among the patients and the controls except for the eosinophil and basophil counts that were skewed. TLC was 2.7–14.9 × 10^9^/L in the patients (mean 6.6 × 10^9^/L) and 2.7–9.1 × 10^9^/L in the controls (mean 5.6 × 10^9^/L) (*p* < 0.001). TLC was elevated (>8.2 × 10^9^/L) in 17% of the patients and 4% of the controls (*p* = 0.050). In the non-smokers, the respective proportions were 17% and 2% (*p* = 0.019). Among the patients, elevated TLC was found in 28% with and 12% without asthma (*p* = 0.108), and in 18% with and 17% without CRS (*p* = 1.000). 

The mean neutrophil count was 3.77 × 10^9^/L in the patients and 3.08 × 10^9^/L (*p* = 0.001) in the controls. It was elevated (>6.20 × 10^9^/L) in 8.1% of the patients and in none of the controls (*p* = 0.055). Among the patients, an elevated neutrophil count was found in 19% of those with asthma and 2% of those without asthma (*p* = 0.003), 9% of those with CRS, and 7% of those without CRS (*p* = 0.714). 

Mean lymphocyte counts were 2.08 × 10^9^/L in the patients and 1.85 × 10^9^/L in the controls (*p* = 0.046), and elevated (>3.50 × 10^9^/L) in 4.1% of the patients and none of the controls (*p* = 0.329). 

There were no other significant differences in blood counts (red blood cell indices, thrombocyte count, basophils, and monocytes) between the patients and the controls (Appendix A).

#### 3.1.2. Other Inflammatory Markers

Concerning the results of FeNO, ESR, total IgE, and CRP, the distributions were skewed. FeNO (only non-smokers included in the analysis) was 2.6–109 ppb among the patients (median 17.0 ppb) and 5.7–60.5 ppb among the controls (median 17.1 ppb, *p* = 0.507). ESR was 2–40 mm/h among the patients (median 6 mm/h) and 2–22 mm/h among the controls (median 5 mm/h, *p* = 0.043). Total IgE was 0–715 kU/L among the patients (median 30 kU/L) and 1–671 kU/L among the controls (median 30 kU/L, *p* = 0.725) (Table 1).

CRP among the patients varied from <1.0 to 29 mg/L (median 1.20 mg/L). 

### 3.2. Allergy Test Results

Positive SPT reactions occurred equally often in the patients and the controls (Table 2). 

### 3.3. Work-Related Stress Level and Laboratory Test Findings

The work-related stress level was significantly more often high in the patients than in the controls (26% vs. 6%, *p* = 0.005). The level of perceived stress was related only to an increased count of blood monocytes in the study patients (*p* = 0.016), not to other blood count results, CRP, FeNO, ESR, or total IgE level (Appendix A).

### 3.4. MCS and Laboratory and Allergy Test Findings 

MCS, defined as high scores in the chemical intolerance subscale, was significantly more common among the patients than among the controls (39% vs. 10%, *p* < 0.001). MCS was not associated with the results of blood count, CRP, FeNO, ESR, or total IgE level (Appendix A). 

Among the 26% of patients presenting with the most difficult MCS symptoms, that is scoring high in all QEESI© subscales (40–100 points in the chemical intolerance and symptom severity subscales, and 24–100 points in the life impact subscale) B-eos was the only laboratory test showing significantly more elevated (>0.30 × 10^9^/L) values when compared to corresponding results of the rest of the patients (23.1% vs. 8.2%, *p* = 0.018) (Appendix A). When the patients with the most difficult MCS symptoms were compared to the rest of the patients, no statistically significant differences in the frequency of asthma (39% vs. 30%, *p* = 0.436), CRS (4% vs. 14%, *p* = 0.279), and allergic sensitization (35% vs. 40%, *p* = 0.645) were observed. 

## 4. Discussion

Among patients referred to secondary health care due to workplace MD associated respiratory and/or voice symptoms, slight elevations in blood leucocyte and neutrophil counts were observed. As previously published, 32% of the patients were diagnosed with asthma and 11% with CRS [28]. A statistically significant relationship between findings of elevated leucocyte or neutrophil counts and diagnosed CRS could not be demonstrated, but among patients with asthma, the elevated neutrophil count was more common than among non-asthmatics. This finding does not necessarily indicate that a large proportion of asthma in these patients has associated with airway neutrophilic inflammation, as it can only be reliably assessed from airway samples [29]. A previous finding among these patients was that 23% of the 30 new-onset asthma cases had signs of type 2 inflammation (increased FeNO and/or levels of blood eosinophils) [28]. These results suggest that MD-associated asthma is less often type 2 asthma as usually seen in adult-onset asthma [30]. Serum total IgE levels did not differ between patients and controls which is in line with previous studies among subjects with workplace MD exposure [7,8].

The levels of CRP and ESR were low, indicating a low probability of inflammatory processes or infections explaining the symptoms [31]. The level of CRP has previously been shown to associate with MD in main living areas at home among children [4]. This discrepancy could be attributed to different immunological responses in children, but CRP remaining low among the patients in this study could suggest that it is the quality of MD exposure at a workplace that fails to induce a systemic inflammatory response. Furthermore, the fact that FeNO and B-eos levels remained normal in this patient group contradictory to previous studies among employees of workplaces with MD [6,7] could indicate exposure differences among the patients. A further study could assess CRP, B-eos, and FeNO levels with respect to the extent and location of MD in relation to symptomatic workers.

The level of perceived stress in the study patients was related to an increased count of monocytes. This result is in line with a previous study by Heidt et al. [32] but probably has little clinical importance. The finding of neutrophilia associated with stress in the study by Heidt et al. was not confirmed in this study. Contradictory to some previous studies [33,34] CRP was not associated with perceived stress in this patient group. 

Earlier in comparison with the general population, MCS was found common in this patient group [35]. In this study when comparing with asymptomatic controls, this finding was confirmed. There was no difference in allergic sensitization between the patients and the controls, and sensitization was not associated with MCS. MCS was also not connected to other laboratory test results even if eosinophil count was associated with the most severe MCS symptoms among the patients. However, this could not be explained by asthma, CRS, or allergic sensitization.

A limitation of the study is that the tests were conducted only at one time point which, on the other hand, does reflect the usual diagnostic measures taken in outpatient clinics. Due to the study settings, the results of this study need to be interpreted judiciously.

## 5. Conclusions

In this study examining workplace MD-exposed patients, there was no basic laboratory or allergy test results characteristic of this patient group. The levels of CRP and ESR were low, and the study patients’ FeNO, total IgE, or allergic sensitization were not increased. Considering that MD-associated symptoms are difficult enough to require examinations in secondary health care, inflammatory processes should still be excluded from basic laboratory tests. However, the use of allergy tests does not seem necessary when the symptoms are clearly workplace-related.

## Figures and Tables

**Table 1 healthcare-11-00971-t001:** Laboratory test results of the study patients and the controls (FeNO = fractional exhaled nitric oxide, ESR = erythrocyte sedimentation rate, IgE = serum total immunoglobulin E).

Laboratory Test	Value/Range	Study Patients % (*n* = 99)	Controls (*n* = 48)	*p*
FeNO ppb	<25 25–50 >50	69.2 26.4 4.4	80.4 15.2 4.3	0.355
ESR mm/h	0–30 >30	98 2.0	100 0	1.000
IgE kU/L	0–100 >100	83.8 16.2	81.3 18.8	0.695

**Table 2 healthcare-11-00971-t002:** Positive reactions to specific allergens in skin prick tests of the study patients and the controls.

	Positive Reactions (%) within Group		
Allergen	Study Patients (*n* = 99)	Controls (*n* = 48)	*p*
1. Birch	20	27	0.401
2. Timothy	23	23	1.000
3. Mugwort	15	21	0.483
4. Horse	5	6	0.716
5. Dog	16	17	1.000
6. Cat	10	21	0.121
*7. Dermatophagoides pteronyssinus*	2	8	0.089
8. Latex	0	0	N.A.
*9. Aspergillus fumigatus*	2	2	1.000

## Data Availability

The data presented in this study are available on request from the corresponding author.

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
