# Peer review of "Laboratory Test Results in Patients with Workplace Moisture Damage Associated Symptoms—The SAMDAW Study"

_healthcare, 2023, doi:10.3390/healthcare11070971_

Round 1

Reviewer 1 Report

In the study entitled “Laboratory test results in patients with workplace moisture damage associated symptoms – the SAMDAW study” the Authors present a brief report on the usefulness of laboratory and allergy tests among patients in the secondary health care with symptoms associated with workplace moisture damage. Authors rationale is worthy of investigation and the study improves and renews the information currently available in the scientific literature.

The subject of the work is of interest and the topic of the manuscript falls within the journal topic, however, the English language needs to be improved and some clarifications are needed in the methods section.

In the abstract section Authors should indicate the statistical analysis applied in the study. Moreover, Authors should better emphasize, in the last sentence of abstract section, the significance of the study.

I suggest to avoid personal form (i.e. our, we, etc.) throughout the manuscript.

In the methods section, Authors should indicate whether exclusion and/or inclusion criteria were selected for the enrolment of subjects in the investigation.

In the statistical analysis paragraph, Authors should indicate whether the normal distribution of data has been assessed.

Results are well written and clear however, discussion section is too short. Authors should better discuss and justify the results obtained in the study.

I suggest to add a separated conclusion section in which Authors should summarize the main findings gathered in their investigation as well as they should emphasize the significance of the study.

Author Response

Comments from Reviewer 1

  1. The subject of the work is of interest and the topic of the manuscript falls within the journal topic, however, the English language needs to be improved and some clarifications are needed in the methods section.

Response: We agree with the reviewer. The manuscript has now undergone English revision.

  1. In the abstract section Authors should indicate the statistical analysis applied in the study. Moreover, Authors should better emphasize, in the last sentence of abstract section, the significance of the study.

Response: The statistical tests used in the analysis have now been indicated and conclusions complemented in the Abstract section.

  1. I suggest to avoid personal form (i.e. our, we, etc.) throughout the manuscript.

Response: We agree with this and have incorporated your suggestion throughout the manuscript.

  1. In the methods section, Authors should indicate whether exclusion and/or inclusion criteria were selected for the enrolment of subjects in the investigation.

Response: Thank you for pointing this out. This information has been added in the Methods section (rows 91-94).

  1. In the statistical analysis paragraph, Authors should indicate whether the normal distribution of data has been assessed.

Response: This information has been added in the Results section (rows 138-139 and 157-158).

  1. Results are well written and clear however, discussion section is too short. Authors should better discuss and justify the results obtained in the study.

Response: Thank you for this suggestion. We have expanded the discussion.

  1. I suggest to add a separated conclusion section in which Authors should summarize the main findings gathered in their investigation as well as they should emphasize the significance of the study.

Response: Thank you for this suggestion. A Conclusions section has been added.

Reviewer 2 Report

Thank you for the  opportunity  to review  this manuscript. In order  to improve its  clinical  quality,  some  issues   must be addressed:

1)Beginning  from the  introduction,  focus  on the clinical repercussion (i.e. if possible focus on  economic/ social impact  surrogate  markers)

2)Focus  your  statistic approach  on  variables  with  both  clinical  and statistical  significance.

3) Aim to perform a multivariate  analysis.

Author Response

Comments from Reviewer 2

1) Beginning from the introduction, focus on the clinical repercussion (i.e. if possible focus on  economic/ social impact  surrogate  markers)

Response: We thank the reviewer for taking the time to assess our manuscript. The focus of this study was to examine the usefulness of laboratory and allergy tests among these patients as a part of their comprehensive clinical evaluation in secondary health care. The results were expected to help targeting the examinations so that they would consist of clinically and scientifically valid tests.

2) Focus your statistic approach on variables with both clinical and statistical significance.

Response: Thank you for pointing this out. The rationale behind analysing many different variables was that this subject has not been studied with this setting before. Also, we see that also negative findings (i.e., without statistical significance) are important to publish.

3) Aim to perform a multivariate analysis.

Response: We appreciate the reviewer’s comment. We did not consider multivariate analysis as a useful method for this research question.

Reviewer 3 Report

In this paper, the authors tried to create a laboratory examination routine for MD patients based on the previous clinical study. The author should answer the following concerns:

1.     Line 102-103: the author should explain why the female ratio is more than 77% in both the patient and control groups. Is there a preference?

2.     The author should clarify how many times the test was conducted for each patient. If it’s multiple times, what’s the time interval?

3. Lines 106-107, the author should include new test data here but not previous data. Previous data can be included in the discussion.

4.     Lines 112-113, the author should clarify what’s the significance to distinguish non-smokers from the overall patient.

5.     Line 113-114; 118-119: What’s the standard to determine whether the patient is asthma or CRS? For example, TLC was found in 28%  with and 12% without asthma, what are the other 60% of people belong to?

6.     What are the chemicals included in MCS evaluation?

Author Response

Comments from Reviewer 3

  1. Line 102-103: the author should explain why the female ratio is more than 77% in both the patient and control groups. Is there a preference?

Response: Thank you for pointing this out. All patients referred because of symptoms associated with suspicion of MD at their workplace were interviewed as possible study participants. The final study population consisted of 99 patients, 82 of whom were women and 17 men. Of the 28 patients who did not take part in the study, 89% were women. The conclusion is that most of the referrals were of women which in turn is probably explained by the fact that women usually have more indoor air quality associated symptoms (Reijula, K.; Sundman-Digert, C. Assessment of indoor air problems at work with a questionnaire. Occup. Environ. Med. 2004, 61, 33–38). On the other hand, In Finland, it is estimated that 20–26% of hospitals and healthcare centers and 12–18% of schools and kindergartens have significant MD, whereas in office buildings the respective proportion is estimated to be 2.5–5%. About 80% of primary-level teachers in Finland are women, and in the Finnish trade union of healthcare employees, 92% of members are women. The high proportion of women in our study is thus at least partly due to more women working in public buildings that have MDs. [1]

Symptomless subjects were recruited with similar proportions of women and men in different age groups as in the study population to ensure that different gender or age distributions would not influence the results.

  1. The author should clarify how many times the test was conducted for each patient. If it’s multiple times, what’s the time interval?

 Response: Thank you for pointing this out. Single tests were used, and this information is added in the Materials and Methods section.

  1. Lines 106-107, the author should include new test data here but not previous data. Previous data can be included in the discussion.

 Response: Thank you for this suggestion. The previous data has been moved to the Discussion section.

  1. Lines 112-113, the author should clarify what’s the significance to distinguish non-smokers from the overall patient.

 Response: This has been clarified.

  1. Line 113-114; 118-119: What’s the standard to determine whether the patient is asthma or CRS? For example, TLC was found in 28%  with and 12% without asthma, what are the other 60% of people belong to?

 Response: Asthma was diagnosed based on symptoms and the demonstration of reversible or variable airway obstruction in lung function measurements: (i) peak expiratory flow (PEF) monitoring, (ii) spirometry with bronchodilation test, or (iii) methacholine challenge test. In the ORL specialist’s clinical evaluation, the diagnostic criteria for chronic rhinosinusitis (CRS) were in accordance with the EPOS2012 guideline, with symptoms of nasal discharge, nasal blockage, hyposmia, facial pressure/pain or nocturnal coughing for

at least 12 weeks and signs of pus in the middle meatus, or pathologic imaging findings in CBCT scans. The patients without asthma and/or CRS were not diagnosed with any illnesses that could explain their workplace associated symptoms. [1]

  1. What are the chemicals included in MCS evaluation?

 Response:  The QEESI© questionnaire was used to explore if the individual has symptoms in association with engine exhaust, tobacco smoke, insecticides, gasoline, paint or thinner, perfumes or fragrancies, fresh asphalt or tar, nail polish, nail polish remover or hairspray, or new furnishings.

[1]       P. Nynäs et al., “Clinical Findings among Patients with Respiratory Symptoms Related to Moisture Damage Exposure at the Workplace—The SAMDAW Study,” Healthcare, vol. 9, no. 9, p. 1112, Aug. 2021.

Round 2

Reviewer 2 Report

More  clinically  oriented,  still  it  is  needed  to  precise the  way normal distribution  of  data was  confirmed,  otherwise it  is  ready  to  be  published.

Author Response

More clinically oriented, still it is needed to precise the way normal distribution  of  data was  confirmed,  otherwise it  is  ready  to  be  published.

Response: We thank the reviewer for pointing this out. The distributions of the parameters were analysed from descriptives (differences between mean and 5% trimmed mean, skewness), Q-Q-plots and histograms. In fact, now re-checked, the distributions of blood eosinophils and basophils were skewed. However, this had been taken into account when testing basophils, but analysing eosinophils with Mann-Whitney test instead of t-test produced a slight change of p value (0.947 à 0.823). This has been changed in the supplementary material.

Reviewer 3 Report

The revised paper should not use reviewed version to show the change tracking.

Author Response

The revised paper should not use reviewed version to show the change tracking.

 Response: We thank the reviewer for the comment. According to the journal instructions, the revised text was returned with tracked changes. We have now attached a revised text with corrections incorporated.
